# Association of Types of Sleep Apnea and Nocturnal Hypoxemia with Atrial Fibrillation in Patients with Hypertrophic Cardiomyopathy

**DOI:** 10.3390/jcm12041347

**Published:** 2023-02-08

**Authors:** Haobo Xu, Juan Wang, Shubin Qiao, Jiansong Yuan, Fenghuan Hu, Weixian Yang, Chao Guo, Xiaoliang Luo, Xin Duan, Shengwen Liu, Rong Liu, Jingang Cui

**Affiliations:** Department of Cardiology, Fuwai Hospital, National Center for Cardiovascular Diseases, Chinese Academy of Medical Sciences, Beijing 100037, China

**Keywords:** hypertrophic cardiomyopathy, obstructive sleep apnea, central sleep apnea, nocturnal hypoxemia, atrial fibrillation

## Abstract

Background: Data regarding the association between sleep apnea (SA) and atrial fibrillation (AF) in hypertrophic cardiomyopathy (HCM) are still limited. We aim to investigate the association of both types of SA, obstructive sleep apnea (OSA) and central sleep apnea (CSA), and nocturnal hypoxemia with AF in HCM. Methods: A total of 606 patients with HCM who underwent sleep evaluations were included. Logistic regression was used to assess the association between sleep disorder and AF. Results: SA was presented in 363 (59.9%) patients, of whom 337 (55.6%) had OSA and 26 (4.3%) had CSA. Patients with SA were older, more often male, had a higher body mass index, and more clinical comorbidities. Prevalence of AF was higher in patients with CSA than patients with OSA and without SA (50.0% versus 24.9% and 12.8%, *p* < 0.001). After adjustment for age, sex, body mass index, hypertension, diabetes mellitus, cigarette use, New York Heart Association class and severity of mitral regurgitation, SA (OR, 1.79; 95% CI, 1.09–2.94) and nocturnal hypoxemia (higher tertile of percentage of total sleep time with oxygen saturation < 90% [OR, 1.81; 95% CI, 1.05–3.12] compared with lower tertile) were significantly associated with AF. The association was much stronger in the CSA group (OR, 3.98; 95% CI, 1.56–10.13) than in OSA group (OR, 1.66; 95% CI, 1.01–2.76). Similar associations were observed when analyses were restricted to persistent/permanent AF. Conclusion: Both types of SA and nocturnal hypoxemia were independently associated with AF. Attention should be paid to the screening of both types of SA in the management of AF in HCM.

## 1. Introduction

Hypertrophic cardiomyopathy (HCM) is the most common hereditary cardiomy-opathy characterized by left ventricular hypertrophy and a spectrum of clinical manifestations [1]. Atrial fibrillation (AF) is the most common cardiac arrhythmia and is associated with significant morbidity and mortality in patients with HCM [2]. Nowadays, considerable evidence supports sleep apnea (SA) as a risk factor for AF [3]. Previous studies including our works showed a high prevalence of SA in HCM and the most common type of SA, obstructive sleep apnea (OSA), is independently associated with AF in HCM [4,5,6]. Unlike OSA, central sleep apnea (CSA) is characterized by a lack of drive to breathe during sleep and is less common [7]. Recently, an increasing number of investigations have linked CSA to AF [8,9]. Whether CSA also has a relationship with AF in HCM is still unknown. In addition, few studies have analyzed the respective relations of OSA and CSA to AF in HCM and there is also a paucity of evidence on the association between nocturnal hypoxemia, an essential pathophysiological feature in SA, and AF. To address the above limitations, the overall aim of the current study was designed to examine the association of both types of SA and nocturnal hypoxemia with AF in a large HCM cohort.

## 2. Materials and Methods

### 2.1. Study population

This retrospective cross-sectional study included consecutive patients who were diagnosed with HCM and underwent overnight diagnostic sleep examination at the inpatient department of Fuwai Hospital between February 2010 and January 2019. The study cohort has been described in detail previously [10]. Diagnostic criteria of HCM were consistent with the 2020 American Heart Association/American College of Cardiology which mainly include unexplained septal hypertrophy with a thickness of at least 15 mm [11]. Patients with rest left ventricular outflow tract (LVOT) peak gradient ≥ 30 mmHg or rest LVOT peak gradient < 30 mmHg with provoked LVOT peak gradient ≥30 mmHg were considered as obstructive. Otherwise, patients were considered as nonobstructive. Patients were excluded if they had New York Heart Association (NYHA) class IV, incomplete sleep data, were younger than 18 years old, had septal reduction therapy before (septal myectomy or alcohol septal ablation), or had a history of heart transplantation. Patients were also excluded if they were receiving treatment with continuous positive airway pressure or oxygen therapy. According to the exclusion criteria, a total of 606 patients were finally enrolled. All patients provided informed consent before enrollment. The study was approved by the ethics committee of Fuwai Hospital (2020-ZX25). All studies were conducted in accordance with the ethical principles stated in the Declaration of Helsinki.

### 2.2. Definition of AF

Prevalence of AF was documented. Data including medical histories, 12-lead elec-trocardiograms and 24-h Holter electrocardiography were collected to help with diagnosis during inpatient stays. Type of AF was defined according to the 2017 HRS/EHRA/ECAS/APHRS/SOLAECE expert consensus statement on catheter and surgical ablation of atrial fibrillation [12]. Briefly, paroxysmal AF was defined as AF that terminated without intervention within 7 days of onset. Persistent AF was defined as continuous AF that is sustained over 7 days. Long-standing persistent AF that lasted at least 1 year when deciding to adopt a rhythm control strategy was classified into persistent AF in our study. Permanent AF was defined when AF was accepted by the patients and physicians and stop further attempts to restore or maintain sinus rhythm.

### 2.3. Diagnosis of Sleep Apnea

Portable polysomnography monitoring was performed before the time of septal reduction therapy by using the system Embletta (Medcare Flaga, Reykjavik, Iceland) in all included patients. All patients underwent testing on room air. This device records nasal airflow by an airflow pressure transducer, finger pulse oximetry, thoracic and abdominal movement, body position, snoring, heart rate, and ECG, and has been validated against full polysomnography [13]. All polysomnograms were scored blindly. Apnea was defined when cessation of airflow or airflow reduction to ≤10% of the baseline value lasted for 10 s or more. An apnea was scored as obstructive if a respiratory effort was present during the event or central in the absence of effort during the event. Hypopnea was defined as a 50% or discernible decrement in airflow lasting 10 s or longer associated with oxygen desaturation of 3%. Hypopneas were scored as obstructive if snoring, and/or flow limitation was noted on the nasal pressure signal or if paradoxical movement was noted on respiratory inductance plethysmography during the event. In the absence of snoring, flow limitation, and paradoxical movement, the hypopnea was scored as a central event. It should be noted that the precise scoring of obstructive or central hypopnea is difficult without the measurement of esophageal pressure. The apnea–hypopnea index (AHI) was the number of apneas and hypopneas per hour of total sleep time. Diagnosis of SA was made solely when the AHI was 5 events/h or more, irrespective of daytime symptoms, which allowed objective evaluation of the disease severity [14]. Patients with SA were grouped into CSA when at least 50% of the disordered breathing events were central (apnea or hypopnea); whereas, if greater than 50% of disordered breathing events were obstructive (apnea or hypopnea), patients were grouped into OSA. The oxygen desaturation index, mean oxygen saturation (SaO_2_), minimal SaO_2_, average pulse frequency and snoring proportion were also recorded. The severity of nocturnal hypoxemia was measured based on percentage of total sleep time (TST) spent with SaO_2_ < 90%. It was assessed as a categorical variable (tertiles with T1 [<0.3%], T2 [≥0.3 to <5.1%] and T3 [≥5.2%]) or as a continuous variable in regression models.

### 2.4. Statistical Analysis

The numeric variables were expressed as mean and standard deviation, and the categorical variables were expressed as number (percentage). Continuous variables were tested for normal distribution with the Kolmogorov–Smirnov test. Comparison of categorical variables was performed using the χ^2^ or Fisher’s exact test, as appropriate. Differences of continuous variables between groups were compared using the Student’s unpaired *t*-test or Mann–Whitney U test, as appropriate. Univariate and multivariate logistic regression analyses were used to determine the association between the presence of both types of SA or severity of nocturnal hypoxemia and prevalence of AF. The results are expressed as odds ratio (OR) and 95% confidence interval (CI). Covariates including age, sex, body mass index (BMI), hypertension, diabetes mellitus, cigarette use, NYHA class, and severity of mitral regurgitation were adjusted. Tests of interaction were performed to assess whether the association of SA and nocturnal hypoxemia with AF was affected by obstruction of LVOT, sex, or obesity using the abovementioned multivariate model. All reported probability values were 2-tailed, and a *p*-value of <0.05 was considered statistically significant. SPSS version 24.0 (IBM Corp., Armonk, NY, USA) was used for calculations and illustrations.

## 3. Results

### 3.1. Baseline Characteristics

A total of 606 patients were enrolled. The study flowchart is shown in Figure 1. SA was diagnosed in 363 (59.9%), of whom 337 (55.6%) had OSA and 26 (4.3%) had CSA. Patients with SA were older, more likely to be male and smokers and had more clinical comorbidities such as hypertension, hyperlipidemia, diabetes mellitus, and coronary heart disease (Table 1). Compared with patients without SA, the prevalence of AF was higher in patients with SA (26.7% versus 12.8%, *p* < 0.001) (Table 1) and the prevalence was even higher in the CSA group when compared with the OSA group (Table 1 and Figure 2A). In addition, patients with CSA had higher NYHA class and N-terminal brain natriuretic peptide (NT-pro BNP) level. Prevalence of AF increased with tertiles of percentage of TST spent with SaO_2_ < 90% (14.9% in T1, 16.8% in T2 and 31.7% in T3, *p* < 0.001) (Figure 2B).

### 3.2. Echocardiographic Data

Echocardiographic data are shown in Table 2. Obstruction of LVOT was more common in patients without SA, while the LVOT gradient in obstructive HCM was not different between groups. Patients with SA were associated with enlarged left atrial diameter (LAD), left ventricular end-diastolic diameter, and ascending aorta diameter compared with patients without SA. LAD was even larger in the CSA group than the OSA group. The mean LVEF was lower and the ratio of patients with LFEV < 50% was higher in the CSA group than the OSA group.

### 3.3. Sleep Parameters

Data from sleep study are summarized in Table 3. Patients with SA had a significantly higher value of AHI and oxygen desaturation index compared with patients without SA. The value was even higher in the CSA group than OSA group. The longest apnea/hypopnea time, percentage of TST spent with SaO_2_ < 90%, and snoring time ratio were higher, and the lowest SaO_2_ and mean SaO_2_ were lower in patients with SA than those without.

### 3.4. Association of Sleep Apnea and Nocturnal Hypoxemia with AF

In the univariate analyses, SA as well as both types of SA, OSA and CSA, were significantly associated with AF (Table 4). OSA was associated with an OR of 3.24 (95% CI, 1.63–6.44), meanwhile, CSA was associated with a higher OR of 6.84 (95% CI, 2.91–16.10). Significant associations were also observed between measures of nocturnal hypoxemia and AF. After controlling for age, sex, BMI, hypertension, diabetes mellitus, cigarette use, NYHA class, and severity of mitral regurgitation, the association of SA (OR, 1.79; 95% CI, 1.09–2.94) and nocturnal hypoxemia (higher tertile of percentage of TST spent with SaO_2_ < 90% with OR, 1.81; 95% CI, 1.05–3.12) with AF remained statistically significant. Adjusted risk of AF was also stronger in the CSA group (OR, 3.98; 95% CI, 1.56–10.13) than the OSA group (OR, 1.66; 95% CI, 1.01–2.76). Similar associations were observed when analyses were restricted to persistent/permanent AF. 

The interaction analysis is shown in Table 5. The association of SA or nocturnal hypoxemia with AF was stronger in obstructive HCM compared with non-obstructive HCM (*p* for interaction = 0.025 and 0.074, respectively). Additionally, associations between SA and AF were greater in obese (BMI ≥ 25 kg/m^2^) patients compared with non-obese (BMI < 25 kg/m^2^) patients (*p* for interaction = 0.024). No significant interaction was present between nocturnal hypoxemia and obesity for AF (*p* for interaction = 0.396). There was no significant interaction between SA or nocturnal hypoxemia and sex for AF (*p* for interaction = 0.534 and 0.793, respectively). 

## 4. Discussion

The current investigation demonstrated that SA was common and was independently associated with AF in patients with HCM. Even though CSA was less common compared with OSA, CSA was more strongly associated with AF than OSA after adjusting for confounders. Nocturnal hypoxemia, an important pathophysiological feature in SA, was also independently associated with AF.

Previous studies have reported a prevalence of SA ranging from 40% to 83% in HCM [15]. Findings from the current study showed that nearly 60% of patients with HCM had SA which was congruent with previous reports, demonstrating that SA was a much more common condition in HCM. The independent association of SA with AF in HCM had been demonstrated in studies with small sample sizes [5,6]. In our study, a significant association between SA and AF was found in a larger HCM cohort. Clinical comorbidities such as obesity, hypertension, and coronary heart disease as well as heart remodeling such as enlarged LAD, which were contributors to AF, were found more common in patients with SA. Several physiologic stressors involved in SA could enhance arrhythmogenicity in HCM including intermittent hypoxemia, hypercapnia, autonomic nervous system fluctuations, and intrathoracic pressure swings. 

However, much of previous data are focused on analyzing the association between OSA and AF, while, the link between CSA and AF is not as well studied. In our study, we first found that both types of SA were independently associated with AF in HCM. Interestingly, CSA was more strongly associated with AF than OSA. These results were in alignment with previous investigations. Sin et al. found AF to be associated with CSA, but not OSA, in a sample of 450 individuals with heart failure [16]. Mehra et al. also documented a stronger cross-sectional relationship of CSA than OSA to AF in an unselected community cohort of 2911 men [17]. Additionally, Tung et al. showed a similar finding that CSA, but not OSA, was a predictor of incident AF in a community-based cohort [9]. CSA may be linked with increased risk for AF beyond OSA through the following mechanisms. Intermittent fluctuations in PaCO2 levels and periodic arousals, occurs to be greater in CSA than OSA, and may predispose to arrhythmia by enhancing sympathetic activation and then resulting in electrical and structural remodeling [18]. CSA is often concomitantly found in patients with systolic heart failure which usually occurred frequently with AF and would exacerbate each other [19]. In our study, the CSA group had higher NYHA class and NT-pro BNP level, enlarged LAD, and decreased LVEF com-pared with the OSA group. These results indicated that the above mechanisms contribute to the stronger association between CSA and AF. Interestingly, only a small proportion of patients with CSA had systolic heart failure. We propose that these patients are more likely to have diastolic dysfunction which is a prominent clinical feature in HCM [20]. 

Nocturnal hypoxemia is an important pathophysiological feature in SA. Previous works had proved the relationship between nocturnal hypoxemia and AF [21]. However, studies have failed to find out the association between nocturnal hypoxemia and AF in HCM, which was possibly due to a relatively small study population [5,6]. In our works, patients in highest tertiles of percentage of TST spent with SaO_2_ < 90% were associated with AF prevalence. We proposed that patients with HCM were associated with AF only with more severe hypoxemia which were consistent with results of previous studies performed in the general population [22]. It is well established that obesity and male sex are risk factors for SA. Our interaction analysis showed that the association between SA and AF was more prominent in patients with obesity but not in the male sex. Meanwhile, the association between nocturnal hypoxemia and AF was irrespective of obesity and sex. Therefore, the association between SA and AF in HCM was not different between males and females. It is still unknown whether LVOT obstruction, a special hemodynamic feature in HCM, plays a role between SA and AF. Our interaction results showed that the association of SA and nocturnal hypoxemia with AF was stronger in HCM patients with LVOT obstruction than those without. These results indicated that obesity and LVOT obstruction might exert synergistic effects on AF together with SA.

To date, SA remains underestimated in HCM. Therefore, a high degree of suspicion for SA is warranted and clinicians should have a low threshold to refer for diagnostic sleep evaluation. Importantly, treatment of OSA was shown to be associated with reduced AF burden as well as cardiovascular benefits in the general population [23,24]. We expect that identification and treatment of both types of SA may serve to further improve outcomes in HCM.

Several limitations in the current study merit discussion. First, this study is a cross-sectional study. Although our results suggested an independent association be-tween SA and AF in HCM, the retrospective nature of this study limited our ability to determine a causal relationship. Second, the sample size of the present study was relatively low, especially in the CSA group, which should be increased in the future to confirm the findings. Third, prevalence of SA and AF in this HCM cohort may represent an overestimation of the prevalence in a general HCM population because of selection bias as patients presented to a tertiary medical center for their care and many were symptomatic. Finally, it is not possible to ensure that all confounding variables were fully adjusted in multivariate analysis. These facts limit the generalizability of our findings.

## 5. Conclusions

Both types of SA and nocturnal hypoxemia were independently associated with AF. Even though CSA was not as popular as OSA in HCM, the association of CSA with AF was stronger than OSA. These results suggest that attention should be paid to the screening of both types of SA in the management of AF in HCM.

## Figures and Tables

**Figure 1 jcm-12-01347-f001:**
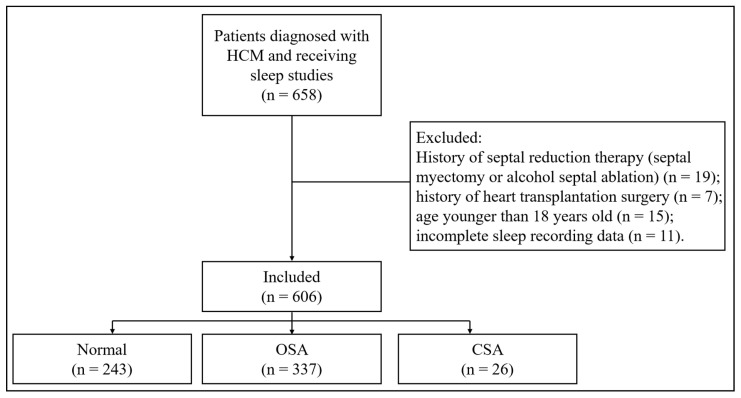
The study flowchart. HCM: hypertrophic cardiomyopathy; OSA: obstructive sleep apnea; CSA: central sleep apnea.

**Figure 2 jcm-12-01347-f002:**
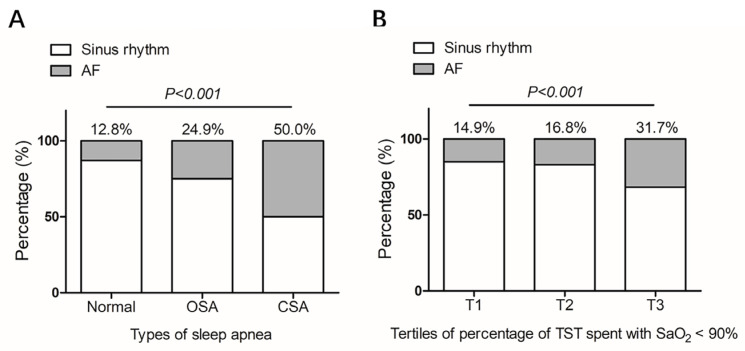
Prevalence of atrial fibrillation in patients with hypertrophic cardiomyopathy grouped by types of sleep apnea (**A**) or severity of nocturnal hypoxemia (**B**). Severity of nocturnal hypoxemia was measured based on percentage of TST spent with SaO_2_ < 90%. Tertiles of percentage of TST spent with SaO_2_ < 90% were classified as following: T1 (<0.3%), T2 (≥0.3 to <5.1%) and T3 (≥5.2%). AF: atrial fibrillation; OSA: obstructive sleep apnea; CSA: central sleep apnea; TST: total sleep time; SaO_2_: oxygen saturation.

**Table 1 jcm-12-01347-t001:** Clinical characteristics of patients with HCM grouped according to types of sleep apnea.

Variables	None (*n* = 243)	SA (*n* = 363)	* *p*-Value	Pooled Patients with SA		# *p*-Value
				OSA (*n* = 337)	CSA (*n* = 26)	
Male	157 (64.6)	278 (76.6)	0.001	258 (76.6)	20 (76.9)	0.966
Age (y)	45.7 ± 13.7	54.0 ± 10.9	<0.001	53.7 ± 11.1	57.2 ± 7.1	0.120
BMI (kg/m2)	24.7 ± 3.3	27.2 ± 3.6	<0.001	27.2 ± 3.6	26.7 ± 3.5	0.501
Cigarette use	92 (37.9)	198 (54,5)	<0.001	181 (53.7)	17 (65.4)	0.249
Hypertension	72 (29.6)	215 (59.2)	<0.001	202 (59.9)	13 (50.0)	0.320
Hyperlipidemia	49 (20.2)	171 (47.1)	<0.001	156 (46.3)	15 (57.7)	0.262
Diabetes	21 (8.6)	63 (17.4)	0.002	56 (16.6)	7 (28.0)	0.148
Coronary heart disease	22 (9.1)	67 (18.5)	0.001	59 (17.5)	8 (32.0)	0.072
History of stroke	9 (3.7)	35 (9.7)	0.006	32 (9.5)	3 (12.0)	0.683
AF	31 (12.8)	97 (26.7)	<0.001	84 (24.9)	13 (50.0)	0.005
Paroxysmal AF	23 (9.5)	55 (15.2)	0.040	50 (14.8)	5 (19.2)	0.547
Persistent or permanent AF	8 (3.3)	42 (11.6)	<0.001	34 (10.1)	8 (30.8)	0.001
NYHA class II–III	154 (63.4)	257 (71.0)	0.049	234 (69.4)	23 (92.0)	0.016
Fasting glucose (mmol/L)	4.8 ± 1.2	5.3 ± 1.5	<0.001	5.3 ± 1.5	5.4 ± 1.3	0.761
TC (mmol/L)	4.4 ± 0.9	4.4 ± 1.0	0.891	4.4 ± 1.0	3.9 ± 1.2	0.009
LDL-C (mmol/L)	2.7 ± 0.8	2.7 ± 0.9	0.810	2.7 ± 0.8	2.4 ± 1.0	0.032
Creatinine (mmol/L)	82.4 ± 15.8	87.4 ± 19.0	0.001	87.2 ± 19.2	90.1 ± 15.4	0.470
NT-pro BNP (pg/mL)	1585.3 ± 2313.9	1276.2 ± 1851.1	0.074	1208.4 ± 1832.8	2188.8 ± 1892.4	0.012

Values are presented as mean ± standard deviation, or as *n* (%). * *p*-value represented comparison between patients with and without sleep apnea. # *p*-value represented comparison between patients with OSA and CSA. HCM: hypertrophic cardiomyopathy; OSA: obstructive sleep apnea; CSA: central sleep apnea; BMI: body mass index; AF: atrial fibrillation; NYHA: New York Heart Association; AF: atrial fibrillation; TC: total cholesterol; LDL-C: low-density lipoprotein cholesterol; NT-pro BNP: N-terminal brain natriuretic peptide.

**Table 2 jcm-12-01347-t002:** Echocardiographic data of HCM patients grouped according to types of sleep apnea.

Variables	None (*n* = 243)	SA (*n* = 363)	* *p*-Value	Pooled Patients with SA		# *p*-Value
				OSA (*n* = 337)	CSA (*n* = 26)	
Obstructive HCM	166 (68.3)	187 (51.5)	<0.001	179 (53.1)	8 (30.8)	0.028
LVOTG at rest in obstructive HCM (mm Hg)	67.3 ± 34.0	63.3 ± 32.4	0.261	62.8 ± 32.5	73.6 ± 31.0	0.358
LAD (mm)	42.1 ± 6.5	43.6 ± 7.7	0.013	43.2 ± 7.5	48.4 ± 7.6	0.001
LVEDD (mm)	43.8 ± 6.3	46.5 ± 5.9	<0.001	46.5 ± 5.9	46.2 ± 6.1	0.825
AAD (mm)	30.2 ± 4.9	32.9 ± 4.3	<0.001	32.9 ± 4.3	33.1 ± 3.8	0.857
IVST (mm)	18.6 ± 5.0	17.2 ± 4.6	0.001	17.1 ± 4.6	18.2 ± 4.2	0.268
PWT (mm)	11.5 ± 3.2	11.6 ± 2.6	0.496	11.6 ± 2.6	11.5 ± 2.1	0.719
Moderate to severe MR	94 (38.7)	104 (28.7)	0.010	96 (28.5)	8 (30.8)	0.804
LVEF (%)	66.6 ± 8.6	65.1 ± 9.4	0.049	65.4 ± 8.9	60.9 ± 13.2	0.016
LVEF < 50%	12 (4.9)	18 (5.0)	0.991	13 (3.9)	5 (19.2)	0.001

Values are presented as mean ± standard deviation, or as *n* (%). The representation of * *p*-value and # *p*-value are shown in Table 1. HCM: hypertrophic cardiomyopathy; OSA: obstructive sleep apnea; CSA: central sleep apnea; LVOTG: left ventricular outflow tract gradient; LAD: left atrial diameter; LVEDD: left ventricular end-diastolic dimension; AAD: ascending aorta diameter; IVST: Interventricular septum thickness; PWT: Posterior wall thickness; MR: mitral regurgitation; LVEF: left ventricular ejection fraction.

**Table 3 jcm-12-01347-t003:** Sleep data of patients with HCM grouped according to types of sleep apnea.

Variables	None (*n* = 243)	SA (*n* = 363)	* *p*-Value	Pooled Patients with SA		# *p*-Value
				OSA (*n* = 337)	CSA (*n* = 26)	
AHI (events/h)	1.8 ± 1.4	22.5 ± 16.7	<0.001	21.8 ± 16.6	31.5 ± 15.0	0.004
OAI (events/h)	0.6 ± 0.7	9.1 ± 10.9	<0.001	9.6 ± 11.2	3.1 ± 2.8	0.003
CAI (events/h)	0.1 ± 0.2	2.2 ± 5.5	<0.001	1.0 ± 2.0	17.1 ± 11.4	<0.001
OHI (events/h)	1.1 ± 1.1	9.9 ± 7.9	<0.001	10.4 ± 7.9	3.4 ± 3.3	<0.001
CHI (events/h)	0.0 ± 0.0	1.3 ± 3.3	<0.001	0.7 ± 2.1	8.1 ± 6.7	<0.001
ODI (events/h)	2.8 ± 2.7	21.1 ± 16.1	<0.001	20.6 ± 15.9	27.4 ± 17.4	0.040
Longest apnea/hypopnea time (s)	41.0 ± 25.1	72.2 ± 25.3	<0.001	72.1 ± 25.6	73.2 ± 7.9	0.839
Lowest SaO_2_ (%)	88.0 ± 4.0	80.2 ± 7.9	<0.001	80.3 ± 7.7	78.6 ± 9.9	0.294
Mean SaO_2_ (%)	94.5 ± 1.7	93.1 ± 2.2	<0.001	93.2 ± 2.1	92.4 ± 2.5	0.080
Percentage of TST spent with SaO_2_ <90% (%)	1.7 ± 6.6	11.6 ± 17.2	<0.001	11.1 ± 16.9	17.6 ± 20.1	0.063
Snoring time ratio (%)	5.1 ± 6.6	11.8 ± 13.6	<0.001	12.0 ± 13.5	9.1 ± 14.8	0.292
Mean HR during sleep (rpm)	59.9 ± 7.3	61.4 ± 7.8	0.019	61.2 ± 7.6	64.4 ± 10.2	0.040
TST (min)	486.7 ± 94.2	456.2 ± 78.7	<0.001	456.4 ± 80.2	453.1 ± 58.3	0.838

Values are presented as mean ± standard deviation. The representation of * *p*-value and # *p*-value are shown in Table 1. HCM: hypertrophic cardiomyopathy; OSA: obstructive sleep apnea; CSA: central sleep apnea; AHI: apnea hypopnea index; OAI: obstructive apnea index; CAI: central apnea index; OHI: obstructive hypopnea index; CHI: central hypopnea index; ODI: oxygen desaturation index; SaO_2_: oxygen saturation; TST: total sleep time; HR: heart rate.

**Table 4 jcm-12-01347-t004:** Association of types of sleep apnea and nocturnal hypoxemia with AF in patients with HCM.

Terms	Univariate	*p*-Value	Multivariate	*p*-Value
Atrial fibrillation				
Sleep apnea	2.49 (1.60–3.88)	<0.001	1.79 (1.09–2.94)	0.022
Types of sleep apnea		<0.001		0.010
None	Reference		Reference	
OSA	2.27 (1.45–3.56)	<0.001	1.66 (1.01–2.76)	0.048
CSA	6.84 (2.91–16.10)	<0.001	3.98 (1.56–10.13)	0.004
Percentage of TST spent with SaO_2_ <90%				
Continuous variable	1.02 (1.01–1.03)	0.001	1.01 (0.99–1.02)	0.257
Tertiles		<0.001		0.012
T1	Reference		Reference	
T2	1.16 (0.68–1.98)	0.586	0.88 (0.50–1.55)	0.652
T3	2.66 (1.63–4.33)	<0.001	1.81 (1.05–3.12)	0.034
ODI	1.01 (1.00–1.03)	0.019	1.00 (0.99–1.02)	0.557
Persistent/permanent atrial fibrillation				
Sleep apnea	3.84 (1.77–8.34)	0.001	2.79 (1.19–6.53)	0.018
Types of sleep apnea		<0.001		0.007
None	Reference		Reference	
OSA	3.30 (1.50–7.25)	0.003	2.47 (1.04–5.88)	0.040
CSA	13.06 (4.39–38.87)	<0.001	7.09 (2.10–23.89)	0.002
Percentage of TST spent with SaO_2_ <90%				
Continuous variable	1.02 (1.01–1.03)	0.009	1.00 (0.99–1.02)	0.642
Tertiles		0.007		0.197
T1	Reference		Reference	
T2	1.10 (0.47–2.55)	0.830	0.86 (0.36–2.08)	0.738
T3	2.68 (1.29–5.56)	0.008	1.65 (0.73–3.69)	0.226
ODI	1.02 (1.01–1.04)	0.006	1.01 (0.99–1.03)	0.221

Data are presented as odds ratio (95% confidence interval). Multivariate analysis was adjusted for age, sex, body mass index, hypertension, diabetes mellitus, cigarette use, NYHA class, and moderate to severe mitral regurgitation. AF: atrial fibrillation; HCM: hypertrophic cardiomyopathy; OSA: obstructive sleep apnea; CSA: central sleep apnea; TST: total sleep time; SaO_2_: oxygen saturation; ODI: oxygen desaturation index; NYHA: New York Heart Association.

**Table 5 jcm-12-01347-t005:** Association of sleep apnea and nocturnal hypoxemia with AF in different subgroups of HCM patients.

Terms	Non-Obstructive	Obstructive	*p* for Interaction	Non-Obese (BMI < 25 kg/m^2^)	Obese (BMI ≥ 25 kg/m^2^)	*p* for Interaction	Male	Female	*p* for Interaction
Sleep apnea	1.10 (0.53–2.28)	2.45 (1.15–5.19)	0.025	1.28 (0.57–2.86)	2.29 (1.17–4.47)	0.024	1.92 (1.03–3.56)	1.82 (0.73–4.55)	0.534
Tertiles of percentage of TST spent with SaO_2_ <90%			0.074			0.396			0.793
T1	Reference	Reference		Reference	Reference		Reference	Reference	
T2	0.57 (0.21–1.29)	1.41 (0.56–3.53)		0.90 (0.36–2.28)	0.78 (0.37–1.66)		0.74 (0.37–1.50)	1.09 (0.39–3.03)	
T3	0.95 (0.44–2.07)	3.45 (1.43–8.33)		1.81 (0.70–4.71)	1.63 (0.81–3.31)		1.62 (0.85–3.10)	2.27 (0.77–6.71)	

Data are presented as odds ratio (95% confidence interval). Tertiles of percentage of TST spent with SaO_2_ <90% were classified as following: T1 (<0.3%), T2 (≥0.3 to <5.1%) and T3 (≥5.2%). Mul-tivariate analysis was adjusted for age, sex, BMI, hypertension, diabetes mellitus, cigarette use, NYHA class, and moderate to severe mitral regurgitation. AF: atrial fibrillation; HCM: hypertrophic cardiomyopathy; TST: total sleep time; SaO_2_: oxygen saturation; BMI: body mass index.

## Data Availability

The data underlying this article will be shared on reasonable request to the corresponding author.

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
