# Peer review of "Association of Types of Sleep Apnea and Nocturnal Hypoxemia with Atrial Fibrillation in Patients with Hypertrophic Cardiomyopathy"

_jcm, 2023, doi:10.3390/jcm12041347_

Round 1

Reviewer 1 Report

Congratulations to all the authors for their work on this article. I would have a few comments about the article. 1. Logistic regression was used to assess the association between sleep disorder and AF. Did you use it for determine the independent risk factor?

2. The results were expressed as mean ± standard deviation, or number (percentage). For mean and standard deviation we use numeric variable, and for percentages we use categorical variable. Please, correct the sentence. 

3. Also, I have a question? Why was the lot divided into tertiles and not quartiles? Why was this decision made?

4. Association of types of sleep apnea and nocturnal hypoxemia with AF in patients with HCM. What statistical test did you use here? Cox regresion or Odd ratio test or maybe Chi-square test? 

Reviewer 2 Report

The Authors present a study of the association of types of sleep apnea and nocturnal hypoxemia with atrial fibrillation in patients with hypertrophic cardiomyopathy.
The manuscript cannot be accepted for publication unless some issues are solved. Please find my comments below.
Major comments
The association between sleep apnea and atrial fibrillation is already known. The Authors should point out, where is the novelty of the presented manuscript.
There is likely a significant selection bias. The Author should explain in detail how the 606 HCM patients were selected for the sleep study.
Why were the patients with septal reduction therapy (SRT) excluded? The Authors report only 19 excluded patients. How do You explain that only 3% of the cohort is treated with SRT? This should be discussed in detail. Compare with the proportion of obstructive patients the presented cohort.
Why were the patients with NYHA class IV excluded?
The Methods section is not sufficient. Is there any follow-up? Is this a cross-sectional study with baseline characteristics only?
It is not clear, why the authors present in the Conclusion that the sleep apnea as "independently associated" with atrial fibrillation. In my opinion, this would need properly conducted multivariable analysis, including variables such as e.g., age, NYHA class, LA diameter, obesity, arterial hypertension, etc. A statistical review is needed here.

Round 2

Reviewer 2 Report

The Authors present a resubmission of a study of the association of types of sleep apnea and nocturnal hypoxemia with atrial fibrillation in patients with hypertrophic cardiomyopathy. Please find my comments below.
Major comments
The association between sleep apnea and atrial fibrillation is already  known. The Authors admitted,  that the results of our study were only supplements to previous works. Therefore, the novelty of the presented manuscript is limited. Consider the scientific priority of this article.
The multivariable analysis did not include LA diameter, as a major predictor of atrial fibrillation.
Table 4. does not include all covariates in the multivariable analysis mentioned in the Authors' rebuttal letter. Was it performed or not?
The study flowchart is not clear. There is enormous selection bias. Only inpatients were included. It also seems that only severely symptomatic patients subsequently recommended for septal reduction therapy were included. Or is it a wrong impression? Methods sections needs to be further updated accordingly.
